# Genetic Features of *Mycobacterium avium* subsp. *paratuberculosis* Strains Circulating in the West of France Deciphered by Whole-Genome Sequencing

Cyril Conde,[a] Julien Thézé,[b] Thierry Cochard,[a] Marie-Noëlle Rossignol,[c] Christine Fourichon,[d] Arnaud Delafosse,[e] Alain Joly,[f] Raphael Guatteo,[d] Laurent Schibler,[g] ◉John P. Bannantine,[h] ◉Franck Biet[a]

[a]INRAE, ISP, Université de Tours, Nouzilly, France
[b]INRAE, EPIA, VetAgro Sup, Université Clermont Auvergne, Saint-Genès-Champanelle, France
[c]INRAE, GABI, AgroParisTech, Université Paris-Saclay, Jouy-en-Josas, France
[d]INRAE, Oniris, BIOEPAR, Nantes, France
[e]Groupement de Défense Sanitaire Orne, Alençon, France
[f]Groupement de Défense Sanitaire Bretagne, Vannes, France
[g]Allice, Paris, France
[h]USDA—Agricultural Research Service, National Animal Disease Center, Ames, Iowa, USA

**ABSTRACT** Paratuberculosis is a chronic infection of the intestine, mainly the ileum, caused by *Mycobacterium avium* subsp. *paratuberculosis* in cattle and other ruminants. This enzootic disease is present worldwide and has a negative impact on the dairy cattle industry. For this subspecies, the current genotyping tools do not provide the needed resolution to investigate the genetic diversity of closely related strains. These limitations can be overcome by the application of whole-genome sequencing (WGS), particularly for clonal populations such as *M. avium* subsp. *paratuberculosis*. The purpose of the present study was to undertake a WGS analysis with a panel of 200 animal field *M. avium* subsp. *paratuberculosis* strains selected based on a previous large-scale longitudinal study of Prim'Holstein and Normande dairy breeds naturally infected with *M. avium* subsp. *paratuberculosis* in the West of France. The pangenome analysis revealed that *M. avium* subsp. *paratuberculosis* has a closed pangenome. The phylogeny, based on alignment of 2,786 nonhomoplasic single nucleotide polymorphisms (SNPs), showed that the strain population is structured into three clades independently of the cattle breed or geographic distribution. The increased resolution of phylogeny obtained by WGS confirmed the homoplasic nature of the markers variable-number tandem repeat (VNTR) and short sequence repeat (SSR) used for *M. avium* subsp. *paratuberculosis* genotyping. These phylogenetic data also revealed independent introductions of the different genotypes in two main waves since at least 2003. WGS applied to this sampling demonstrated the presence of mixed infections in herds and at the individual animal level. Collectively, the phylogeny results inferred with French isolates compared to *M. avium* subsp. *paratuberculosis* isolates from around the world suggest introductions of *M. avium* subsp. *paratuberculosis* genotypes through the animal trade. Relationships between genetic traits and epidemiological data can now be investigated to better understand transmission dynamics of the disease.

**IMPORTANCE** *Mycobacterium avium* subsp. *paratuberculosis* causes Johne's disease in ruminants, which is present worldwide and has significant negative impacts on the dairy cattle industry and animal welfare. Prevention and control of *M. avium* subsp. *paratuberculosis* infection are hampered by knowledge gaps in strain virulence, genotype distribution, and transmission dynamics. This work has revealed new insights into *M. avium* subsp. *paratuberculosis* strains currently circulating in western France and how they are related to strains circulating globally. We applied whole-genome

Address correspondence to Franck Biet, franck.biet@inrae.fr.
The authors declare no conflict of interest.

sequencing (WGS) to obtain comprehensive information on genome evolution and discrimination of closely related strains. This approach revealed the history of *M. avium* subsp. *paratuberculosis* infection in France, refined the pangenomic characteristics of *M. avium* subsp. *paratuberculosis*, and demonstrated the existence of mixed infection in animals. Finally, this study identified predominant genotypes, which allow a better understanding of disease transmission dynamics. This information will facilitate tracking of this pathogen on farms and across agricultural regions, thus informing transmission pathways and disease control points.

**KEYWORDS** *Mycobacterium avium* subsp. *paratuberculosis*, pangenome, phylogeny, whole-genome sequencing, genomics

Paratuberculosis, also called Johne's disease (JD), is one of the most common infectious diseases in livestock worldwide and is associated with a high health impact and substantial socioeconomic costs for governments and farming industries (1). Despite decades of costly surveillance and control campaigns, with expenditures over $250 million annually in the United States (2) and in the same proportion in most industrialized countries, JD remains endemic and is even progressing in certain countries (3). The prevalence of JD remains at very high: around 50% for European herds and around 80% in the United States. In France, the herd prevalence of JD is over 50% for cattle herds and around 63% for goat herds (4). The inability to control this disease, which is characterized by a very slow chronic course, can be explained by important knowledge gaps in both prophylaxis and the causative agent, *Mycobacterium avium* subsp. *paratuberculosis* (5, 6). The infection is particularly slow in development and usually occurs via the fecal-oral route, although infections *in utero* also occur. JD can present in a wide variety of subclinical and clinical forms over an incubation period that can range from 1 to 14 years. Typically, 10% to 15% of infected cattle develop clinical signs. This chronic infection of the intestine, mainly of the ileum, results in discontinuous *M. avium* subsp. *paratuberculosis* excretion, especially in the early stages of infection (7). The risk of an animal becoming infectious (by shedding bacteria in its feces) is a primary concern to the farmer because it leads to disease transmission. The prevention and control of *M. avium* subsp. *paratuberculosis* infection are also hampered by a lack of knowledge on differences in strain virulence, distribution of genotypes, and dynamics of transmission (6). Thus, having an in-depth knowledge of *M. avium* subsp. *paratuberculosis* strain genetics and knowing the associated epidemiological information can bridge this knowledge gap.

Since the use of restriction fragment length polymorphism (RFLP) based on IS*900* (8), the genotyping of strains has been more accessible by molecular biology techniques based on polymorphism of minisatellites or microsatellites, multilocus variable-number tandem repeat (VNTR) analysis (MLVA) typing (using VNTR markers), and multilocus short sequence repeat (MLSSR) typing, respectively (9, 10). However, these techniques, although they are still widely used for the typing of *M. avium* subsp. *paratuberculosis* strains, lack resolution for fine tracing of the *M. avium* subsp. *paratuberculosis* strains circulating between and within herds. *M. avium* subsp. *paratuberculosis* genomic studies initiated in 2016 (11, 12) confirmed the clonal nature of this *M. avium* subspecies, which is divided into two lineages designated type C and type S. The type C strains, commonly isolated from cattle, contain a subgroup of type B strains isolated from cattle and bison. The type S strains, commonly isolated from sheep, are subdivided into subtypes I and III (13).

Through a complete genome analysis of *M. avium* subsp. *paratuberculosis*, we recently showed that type C strains are very stable, with a low number of single nucleotide polymorphisms (SNPs) and accessory genes and a lack of genomic rearrangements (14). This is congruent with the limits observed in the typing methods used alone or in combination (15). Whole-genome sequencing (WGS), with an ultimate resolutive power, makes it possible to know the very precise phylogenetic links between strains in order to infer the population structure and to study associations between genotype and epidemiological data. A similar approach was recently used to examine 197 Irish strains of *M. avium* subsp. *paratuberculosis* (16).

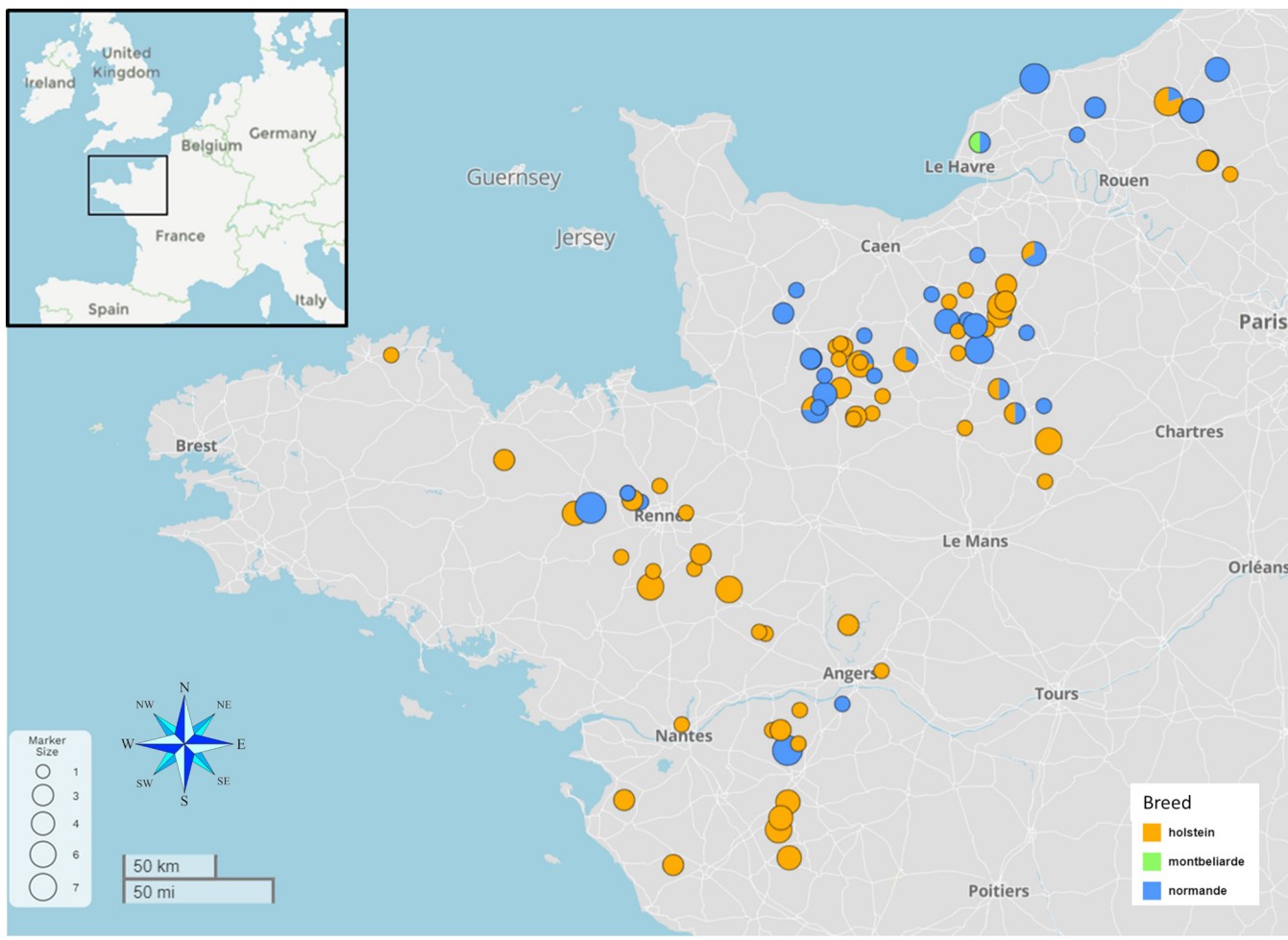

**FIG 1** Geographic locations of collected *M. avium* subsp. *paratuberculosis* samples. Shown is the enlarged section of western France taken from the inset at the top left. The colors correspond to the cattle breed from which the fecal samples were harvested. Circle sizes represent the number of samples collected in the herd. The map was produced using MicroReact (v.202) (76).

In France, the highest concentrations of dairy herds are in western France (17). A national JD program in France made it possible to carry out a vast longitudinal study of *M. avium* subsp. *paratuberculosis* fecal shedding and serological patterns in dairy cattle over 4 years (18). In total, 1,644 Holstein and 649 Normande cows with relevant and accurate contrasted phenotypic profiles (control and nonclinical/clinical cases) were genotyped with a medium-density BovineSNP50 BeadChip and analyzed in a genome-wide association study (GWAS) that identified candidate variants located in genes that were functionally related to resistance to *M. avium* subsp. *paratuberculosis* (19). This program was also an opportunity to establish a large collection of *M. avium* subsp. *paratuberculosis* strains isolated from clinical JD cases in Holstein and Normande cows.

The present work was designed to leverage these previous phenotypic studies and GWASs by conducting a whole-genome analysis on a representative panel of *M. avium* subsp. *paratuberculosis* field strains circulating in western France (Fig. 1). The WGS analysis identified accurate phylogenetic relationships between clinical isolates and other worldwide isolates to ascertain their genomic characteristic and to establish correlations between genomic traits and epidemiological data within a population of well-documented strains.

## RESULTS

**Pangenome of *M. avium* subsp. *paratuberculosis* isolates and comparison to the *M. avium* subsp. *hominissuis* pangenome.** To investigate the genomic structure and gene content of *M. avium* subsp. *paratuberculosis* isolates, we performed a pangenome

analysis. Of the 182 genomes of field *M. avium* subsp. *paratuberculosis* strains circulating in France (Fig. 1; see Data Set 1 in Table S1 in the supplemental material) (20), 137 were selected for pangenome analysis. Genomes with less than 20× coverage or whose assembly yielded more than 400 contigs or <69.2% GC content were excluded from the analysis. For genome assembly, as described in Materials and Methods, the number of contigs ranged from 123 to 358, with an average of 178, and $N_{50}$ ranged from 30,069 to 97,349, with an average of 67,901. The metadata and all sequencing and assembly metrics for all isolates are presented in Table S1, Data Set 1. In addition, we included 4 new fully assembled genomes belonging to the main French clades (see the sections on phylogeny below) (Table S1, Data Set 1), 8 complete *M. avium* subsp. *paratuberculosis* genomes of the bovine type (subtype II), and 4 genomes of subtype B, available at NCBI and detailed in Table S1, Data Set 3. These genomes isolated from around the world were added to include diversity in this analysis. The *M. avium* subsp. *paratuberculosis* gene distribution detected by Panaroo initially revealed a pangenome of 4,564 genes. Most of the genes classified in the shell and cloud genome were annotated as "hypothetical protein" and were not found in the complete genomes (Table S2). As fragmented assemblies were included in this analysis, we aligned the consensus sequences of all gene clusters of the pangenome to the circular genome of strain FDAARGOS_305 with the objective of identifying redundant annotations. These redundant gene clusters correspond to different problems related to the annotation of fragmented genomes, such as (i) differences in the annotation of some genes (nonhomogeneous annotation), (ii) annotation of several genes in place of a single large gene due to genome fragmentation (examples include nonribosomal peptide synthetase or polyketide synthase genes), and (iii) exclusion of some genes present partially at the end of contigs and not annotated. We have identified and removed 123 redundant gene clusters, giving a final pangenome of 4,441 genes, including 4,386 genes (i.e., 98% of genes) in the core genome and 55 genes in the accessory genome (Fig. 2a and b). Among the 123 redundant gene clusters, 5 belong to the core genome (4.07%) and 118 belong to the accessory genome, including 28 soft-core genes (22.76%), 37 shell genes (30.08%), and 53 cloud genes (43.09%). Also, 85 of these genes were annotated as hypothetical protein genes. A BLAST database was constructed containing the nucleotide sequences of the 153 genomes included in the analysis to verify the presence of the remaining 55 accessory genes. Searches against this database revealed that the majority of genes are present partially at the ends of assembled contigs in draft genomes. These analyses confirm that *M. avium* subsp. *paratuberculosis* has high genetic stability regardless of time and country of isolation. For comparison, we repeated the analysis using 114 genomes of *M. avium* subsp. *hominissuis* (Table S1, Data Set 5), the ancestral species of *M. avium* subsp. *paratuberculosis*, with an average nucleotide identity (ANI) value measured between the *M. avium* subsp. *hominissuis* and *M. avium* subsp. *paratuberculosis* strains of >98.6%. It should be noted that on a pangenome of 9,201 gene clusters for *M. avium* subsp. *hominissuis*, the contributions of the core and accessory genomes represent 40.9% (3,763 genes) and 59.1% (5,438 genes), respectively. In contrast to *M. avium* subsp. *paratuberculosis*, although some genes are found at the end of the contigs, many of these genes appear to be true accessory genes because they are found at a distance from the ends of the contigs.

We applied Heaps' law to *M. avium* subsp. *paratuberculosis* to better characterize the breadth of the gene repertoire accessible and the amount of additional genomic data required for characterization of this repertoire. The gene accumulation curves carried out in accordance with Heaps' law (21, 22) (see Materials and Methods) are shown in Fig. 2c. Remarkably, we observed that the core genes and pangenome curves are almost parallel. This result reflects the fact that *M. avium* subsp. *paratuberculosis* has a stable genome. Indeed, the Heap's power-law constant predicted a closed pangenome ($\alpha = 1.85$). Also, during this analysis, out of 1,000 permutations, only 12 new genes on average were added to the pangenome. Conversely, when using 114 *M. avium* subsp. *hominissuis* genomes (Table S1, Data Set 5), an $\alpha$ value of 0.89 is obtained, which is indicative of an open pangenome for *M. avium* subsp. *hominissuis*, showing gene accumulation

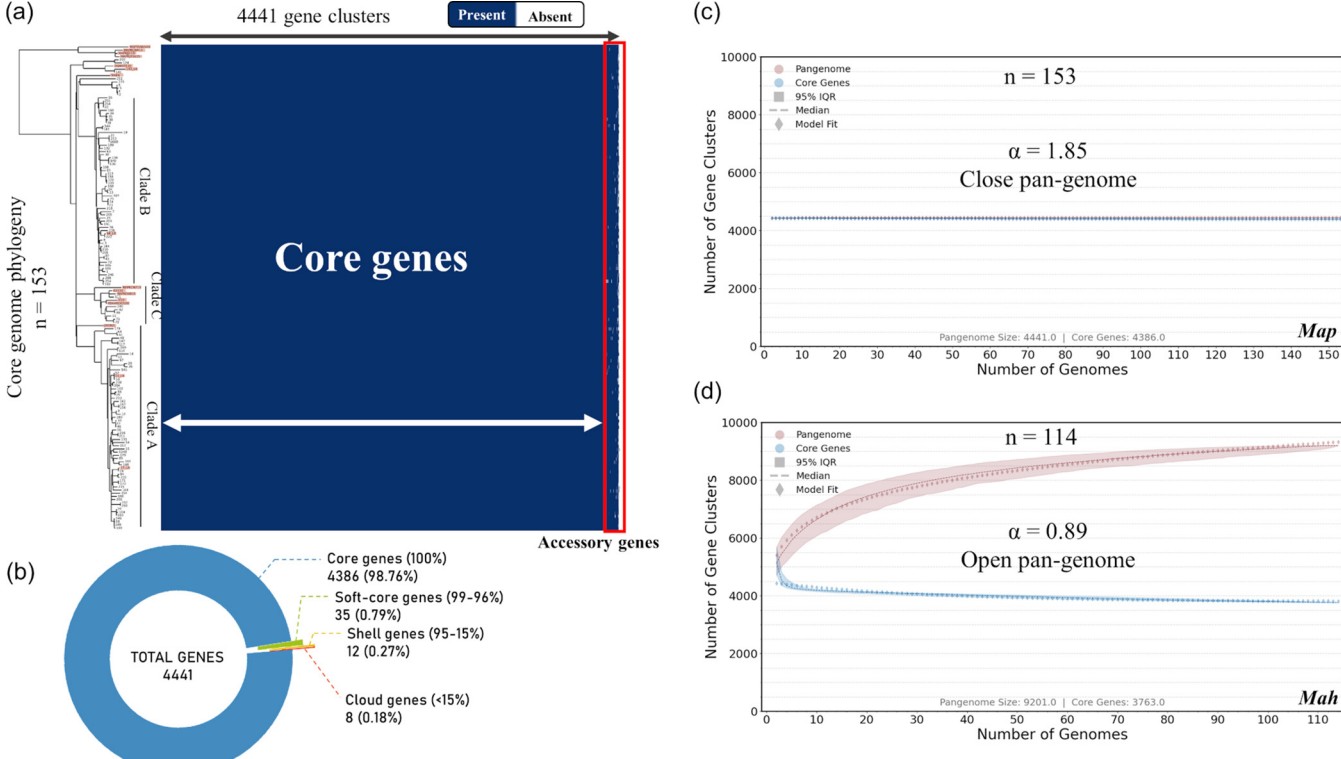

**FIG 2** Pangenome analysis of *M. avium* subsp. *paratuberculosis* (*Map*). (a) Pangenome gene cluster presence/absence matrix of 153 *M. avium* subsp. *paratuberculosis* genomes where blue blocks indicate presence of the gene in that cluster and white indicates its absence. The corresponding maximum likelihood phylogenetic tree, inferred on concatenation of 3,335 core genes (see Materials and Methods) from 153 *M. avium* subsp. *paratuberculosis* genomes, is presented on the left, and isolates listed on the tree correspond to each row of the matrix. Complete genomes are highlighted in orange on the tree. Accessory genes are shown in the red box at the right. (b) Pie chart representing proportion of core genes (present in ≤99% of genomes), shell genes (present in between 15% and 99% of genomes), and cloud genes (present in ≤15% of genomes). The majority of gene clusters belonging to the shell and cloud genomes were annotated as hypothetical proteins. The right half of the figure shows two gene cluster accumulation curves for the pangenome (red) and core genome (blue), with 1,000 isolate permutations for 153 *M. avium* subsp. *paratuberculosis* genomes (c) and 114 *M. avium* subsp. *hominissuis* (*Mah*) genomes (d). The Heap's law α value was determined using the median values of 1,000 strain permutations. Openness of the pangenome is also indicated.

curves with asymptotic behaviors (Fig. 2d). During this analysis, out of 1,000 permutations, 4,460 new genes on average were added to the *M. avium* subsp. *hominissuis* pangenome. Finally, the pairwise homoplasy index (PHI) test did not indicate statistically significant evidence of recombination ($P = 0.41$) in the *M. avium* subsp. *paratuberculosis* core genes' alignment.

**Complete genome sequencing of selected French isolates.** There are currently 10 complete genome sequences for the C-type *M. avium* subsp. *paratuberculosis* of subtype II (bovine type), but none are from France. In order to have a complete French reference genome, necessary for this study, we have chosen four strains distributed in each clade (see phylogeny sections) for sequencing and assembly into a single circular chromosome ranging from 4,837,630 to 4,838,134 bp with a GC content of 69.3%. Alignment of the four genomes with respect to the reference strain K-10 shows that the genomic organization is remarkably stable, with only one large inversion shared among the four strains, corresponding to a previous observation by Talaat and co-workers (23) (Fig. S1).

**Whole-genome SNP phylogeny.** A maximum likelihood (ML) tree was reconstructed using an alignment of 2,786 nonhomoplasic SNPs, including 193 *M. avium* subsp. *paratuberculosis* isolates (the 192 French isolates in Data Sets 1 and 2 and the K-10 genome in Data Set 4 in Table S1). The ML phylogenetic tree (Fig. 3) shows two dominant well-supported monophyletic lineages (100% bootstrap support at basal nodes) and a minor well-supported monophyletic lineage, called clades A, B, and C for clarity only. This population structure was confirmed by fastBAPS analysis. Clades A, B,

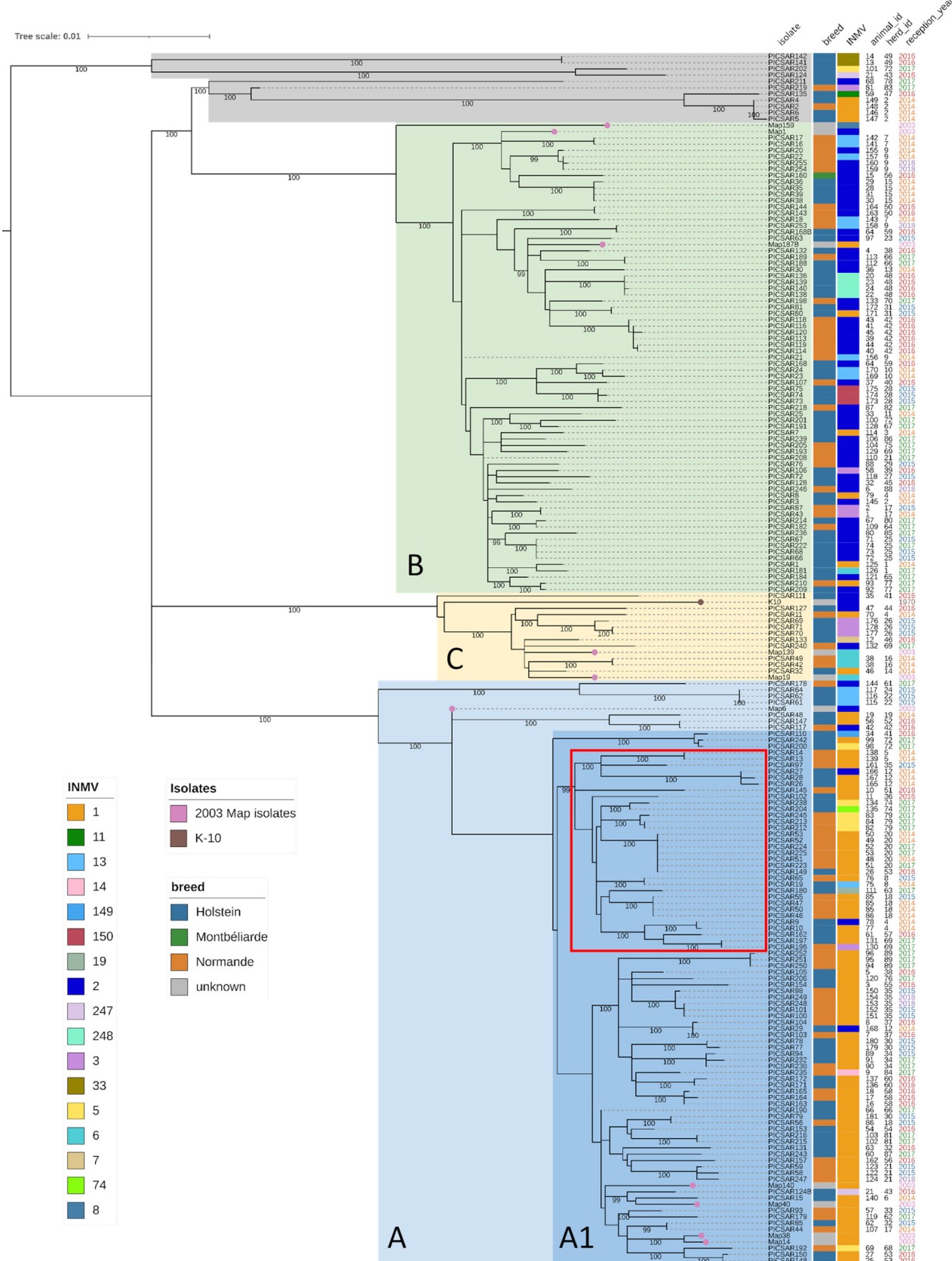

**FIG 3** Phylogenetic lineage of *M. avium* subsp. *paratuberculosis* strains from France. A maximum likelihood phylogenetic tree was inferred using IQ-Tree (K3Pu+F model), showing the relatedness of 192 French isolates and the K-10 strain. PICSAR77 was used as the reference, being completely assembled

**TABLE 1** Neutrality tests used in this study[a]

| Population (n) | Result by: | | | | | | | |
| | Fu's test | | Tajima's test | | Fu's and Li's test | | | |
| | $F_s$ | $P^b$ | D | P | D* | P | F* | P |
|---|---|---|---|---|---|---|---|---|
| Clade A (93) | −18.89 | 0.0028 | −2.715 | 0 | −4.462 | 0.005 | −4.251 | 0.0025 |
| Clade A1 (85) | −15.234 | 0.0069 | −2.631 | 0 | −3.937 | 0.0064 | −3.872 | 0.0039 |
| Clade B (75) | −6.598 | 0.0819 | −2.639 | 0 | −5.345 | 0.0018 | −4.814 | 0.0011 |
| Clade C (14) | 1.294 | 0.6979 | −1.938 | 0.0126 | −2.309 | 0.0235 | −2.31 | 0.0205 |

[a]The P value for each population assumes no recombination. P values were computed based on 10,000 coalescent simulation runs.
[b]The statistic should be considered significant at the 5% level if the P value is <0.02.

and C include 93, 75, and 14 isolates, respectively. These clades are characterized by long branches at basal nodes, high branching rates at internal nodes, and a very low genetic diversity at external nodes (usually reflecting samples isolated from the same herd). This suggests a rapid diversification of *M. avium* subsp. *paratuberculosis* strains from divergent genetic backgrounds followed by population stabilization within these clades. The clade A1 was defined by removing nodes basally anchored in clade A, which reflects other introductions that do not appear to have spread in France. Of note, isolates from clade C were phylogenetically close to reference strain K-10. The remaining isolates form two well-supported monophyletic lineages, including only four and six isolates, with one lineage being basal to the whole phylogeny and the other basal to clade B. These isolates are more deeply anchored in the phylogeny, suggesting a different evolutionary history from the three main clades. The French isolates recovered from 2003 fall into the three main French lineages (five in clade A, three in clade B, and two in clade C), suggesting clades A, B, and C were already circulating in France in 2003 and possibly before.

As the star-like topology of the 3 clades A, B, and C suggests population growth, we performed neutrality tests to provide evidence for these expansions. The neutrality tests in Table 1 show evidence indicating population growth for the population of clade A with significantly negative values for all 3 tests. However, although the Tajima's D and Fu and Li's D* and F* tests were significantly negative for the clade B population, Fu's $F_s$ test did not reach statistical significance. Interestingly, Fu's $F_s$ test was positive for the clade C population, potentially indicating a recent bottleneck. Due to the low number of isolates in the clade C population, this test was not statistically significant.

In total, 156 SNPs (Table S3) discriminate between clades A, B and C, of which 49 SNPs were clade A specific, 45 SNPs were clade B specific, and 62 SNPs were clade C specific. Among the 156 SNPs, 97 were nonsynonymous, 49 were synonymous, and 10 were intergenic. The difference between clades as determined by pairwise SNP distance is shown in Fig. 4. (Supporting data are presented in Table S4.) The heat map shows that isolates within the same clade were separated on average by 54 SNPs (ranging from 0 to 166 SNPs). The distance between 2 isolates from 2 different clades was on average 208 SNPs (ranging from 67 to 245 SNPs). The heat map shows that isolates shaded in gray dots were more distant from other clades, with an average of 221 SNPs (ranging from 11 to 308 SNPs). This wide range was due to the fact that one isolate (i.e., PICSAR219) was close to clade A isolates, with an average of 38 SNPs (ranging from 11 to 88 SNPs). These isolates were also very distant from each other, with an

**FIG 3** Legend (Continued)
and belonging to the largest clade of the phylogeny (i.e., clade A), and homoplasic sites were removed. The tree was midpoint rooted. The scale of the tree is by site substitution. Bootstrap phylogenetic clades have been highlighted (light blue, clade A; dark blue, clade A1; green, clade B; yellow, clade C). The other strains have been shaded in gray. From left to right are shown the isolate name, host breed, MLVA-INMV (INRA Nouzilly MIRU VNTR) profiles, animal identifier, herd identifier, and year of strain isolation. The legends for breed and MLVA profile are shown in the lower left. Colored dots on the tree indicate *M. avium* subsp. *paratuberculosis* isolates from 2003 (pink) or the K-10 strain (brown). The monophyletic group carrying the synonymous mutation in the *gyrB* gene at position 1124 is indicated by a red box.

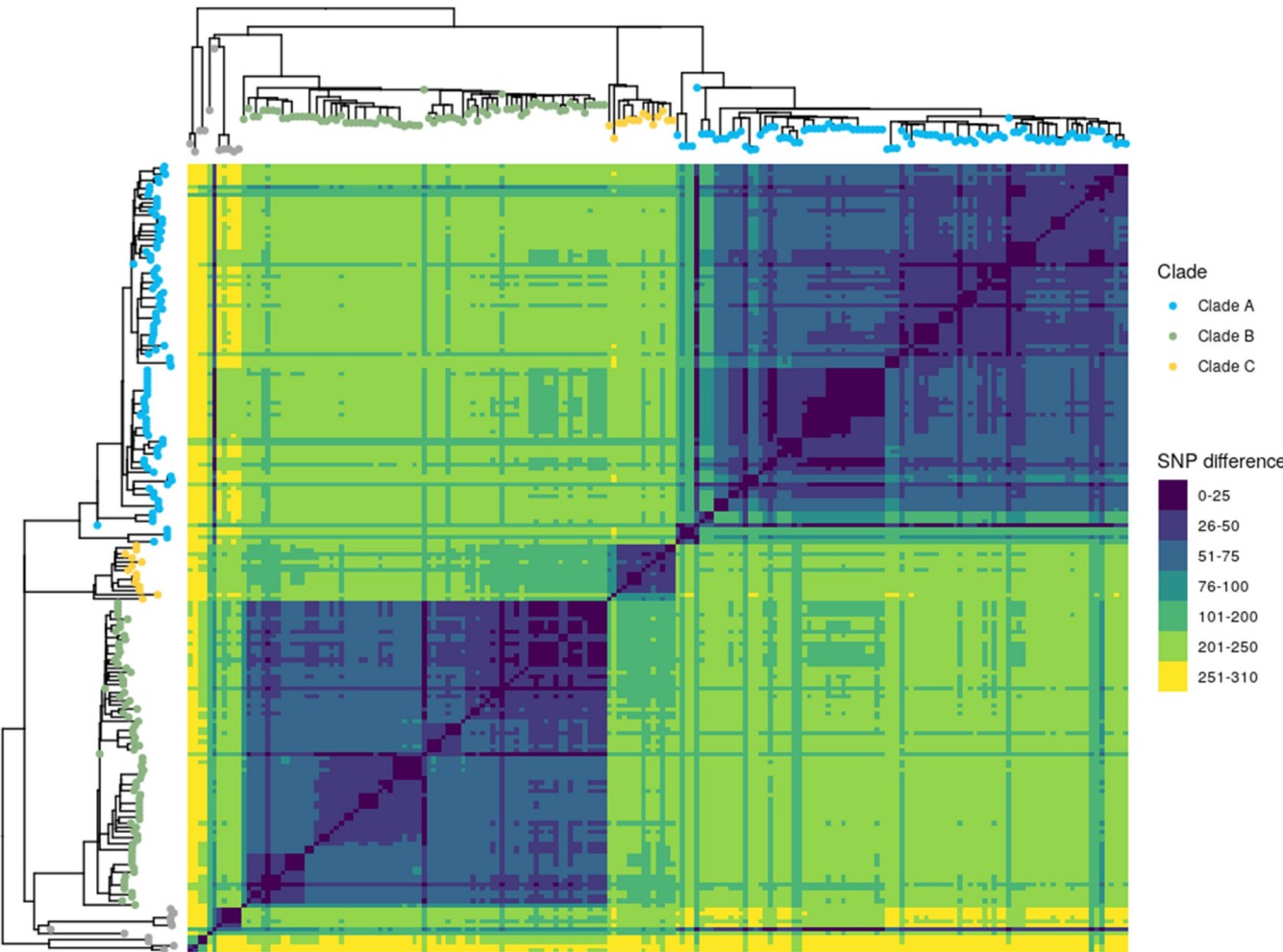

**FIG 4** Closely related *M. avium* subsp. *paratuberculosis* isolates separated into clades. Shown is a heat map illustrating pairwise SNP distances between genomes belonging to each clade. Both axes have a maximum likelihood SNP-based tree inferred on 193 genomes with the leaf colored according to clade. Trees were midpoint rooted. The SNP difference key is shown to the right.

average of 187 SNPs (ranging from 0 to 308 SNPs). These SNPs can now be used to improve epidemiological surveillance in order to better track the circulation of *M. avium* subsp. *paratuberculosis* in western France, which closes a knowledge gap that hindered the control of paratuberculosis. Interestingly we detected a nonsynonymous mutation in the *gyrB* gene at position 1124 in one monophyletic group of isolates within clade A1 (indicated by a red box in Fig. 3).

With the high resolution obtained from whole-genome phylogeny, we were able to identify various cases of infection at the herd and animal levels (summarized in Fig. 5 and Table S4). At the herd level, we observed that multiple *M. avium* subsp. *paratuberculosis* isolates coexisted simultaneously—as, for example, in herd 4, where isolates belonging to clade A (PICSAR9 and PICSAR10), clade B (PICSAR8), and clade C (PICSAR11) were sampled on different animals in the same year (Table S4, Herds_and_Animals tab). In contrast, we identified herds in which the same isolate was sampled from different animals over a 3-year period (herd 20). At the animal level, the same phenomenon (i.e., clonal infection) was observed with 3 identical isolates sampled over a 2-month period from animal 85 (Table S4, animal 85, herd 18). For the first time, we observed animals with mixed infections (infected with different isolates [Table S4, animals 21 from herd 43, 64 from herd 59, and 86 from herd 18]). Concerning animals 21 and 64, two independent cultures of each were propagated from distinct colonies from one primary fecal culture on a slope. In animal 21, the sequencing revealed that the 2 strains isolated differed by 292 SNPs and belonged to two

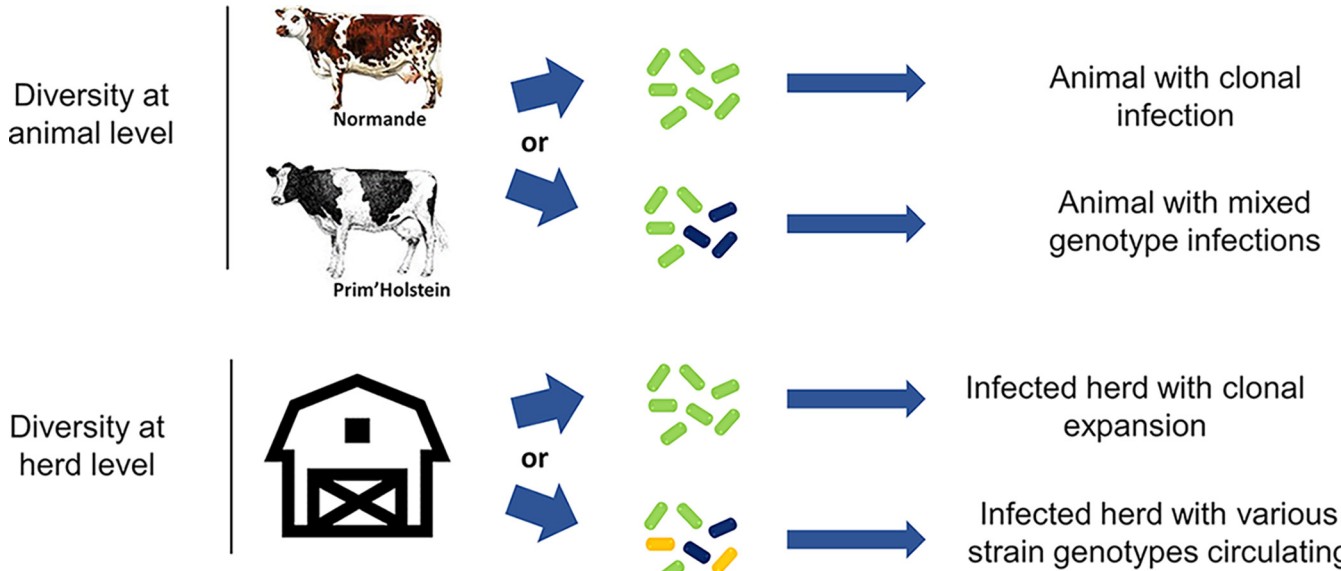

**FIG 5** Infection occurrences observed on dairy farms in France. A schematic illustration shows the types of infection observed at the animal level (top) and the herd level (bottom) in western France.

different clades. In animal 64, the 2 strains isolated differed by 67 SNPs. Two separate fecal samples from animal 86 were obtained 2 months apart. Subculture and analysis of these samples revealed that the 2 *M. avium* subsp. *paratuberculosis* strains isolated differed by 44 SNPs. At no time did we observe a single animal infected with three or more distinguishable genotypes. We considered that the strains were closely related when the SNP counts were between 1 and 100, while strains not directly related contained at least 100 SNPs. Other results exemplified the highly clonal nature of *M. avium* subsp. *paratuberculosis*. For example, in animals 48 and 51 from herd 20, which were sampled several times over a period of 2 to 4 years, WGS analysis showed identical strains (Table S4).

**Association between the whole-genome SNP phylogeny and different traits.** A phylogeny-trait association analysis was performed between the whole-genome SNP Bayesian phylogeny and three different traits: animal breed, shedding score, and MLVA profiles (Table S1). Here, we considered the topological uncertainty of the SNP phylogeny using the set of posterior trees from the Bayesian phylogenetic analysis. Each trait was also tested against the null hypothesis that trait is randomly distributed in the phylogeny. No association (Table S5) could be established between the phylogeny and the animal breed (association index [AI], $P = 0.01$; parsimony score [PS], $P = 0.12$) or the shedding score (AI, $P < 0.01$; PS, $P = 0.04$). On the contrary, a strong statistical association was found between the phylogeny and the MLVA typing (AI, $P < 0.01$; PS, $P < 0.01$). Moreover, the monophyletic clade size (MC) statistic, which quantifies the strength of an association between a state of a trait and the phylogeny, shows a strong association ($P = 0.01$) between the INMV1 profile and the phylogeny. Indeed, as shown in Fig. S2, clade A is primarily associated with the INMV1 profile (representing 76.34% of clade A isolates). Despite the INMV2 profile predominance with clade B (representing 65.33% of clade B isolates), the MC statistic does not show a significant association (MC, $P = 0.10$) with the phylogeny. The results of MLSSR typing did not show any significant correlation with clades defined by phylogeny.

These results have also confirmed the homoplastic nature of these markers since several strains were shown to be identical in sequence but exhibited different MLVA and MLSSR profiles, and conversely, we showed examples of strains separated by several hundred SNPs with identical MLVA and MLSSR profiles.

**Correlation between pairwise phylogenetic and geographical distances.** We investigated whether an association existed between phylogenetic distance and geographic distance of isolates in order to investigate the spread of isolates between

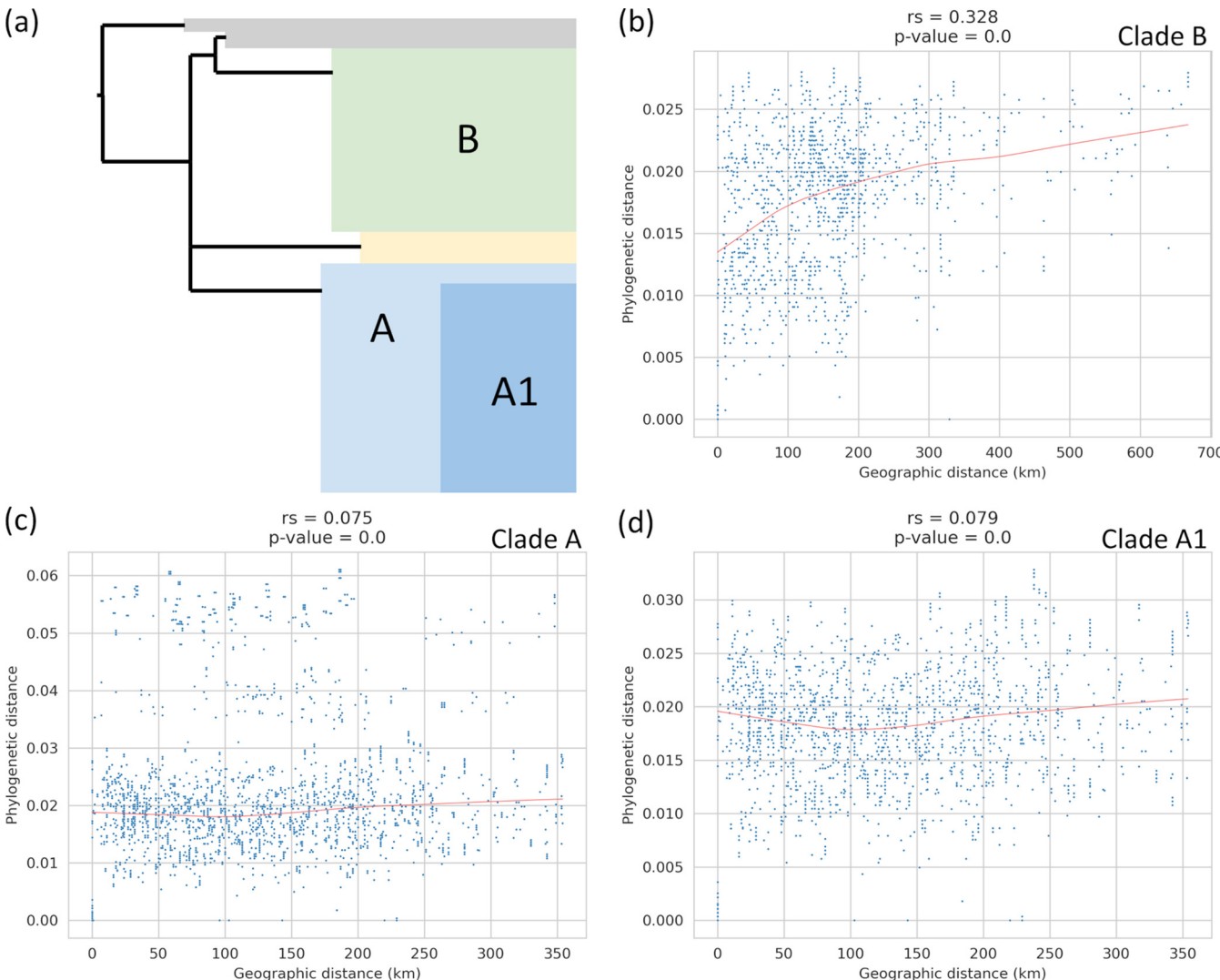

**FIG 6** Correlation between pairwise phylogenetic and geographical distances. Comparison of pairwise phylogenetic distances and pairwise geographical distances obtained from the French isolates in this study using whole-genome SNP analysis. (a) Schematic representation of the topology structure of the ML tree. Correlation was obtained from isolates with redundancy removed in (b) clade B, (c) clade A, and (d) clade A1.

herds. We plotted the pairwise distance in the tree against the pairwise geographic distance between herds (Fig. 6). Redundant isolates were removed and the analysis was performed on each clade independently. We did not observe any correlation between phylogenetic and geographic distance within clade A (Spearman correlation $r_s = 0.075$; $P < 0.01$) (Fig. 6c). However, isolates belonging to basal lineages of clade A seem to decrease the signal (Fig. 6c). We removed these isolates from clade A (clade A1) but still did not observe any correlation between phylogenetic and geographic distances within clade A1 (Spearman correlation $r_s = 0.079$; $P < 0.01$) (Fig. 6d). A weak but statistically significant positive correlation was observed for all clade B (Spearman correlation $r_s = 0.328$; $P < 0.01$) (Fig. 6b). These results suggest that longer geographical distances corelate with strain phylogenetic differences for clade B but not A. However, regarding our sampling, only 12 animals were subjected to movements of distances less than 50 km and only 1 animal was moved 600 km.

**Phylogeny of French isolates in the global context.** Genomic studies on collections of *M. avium* subsp. *paratuberculosis* isolates have recently increased, and data are available publicly. It was therefore of interest to include these data to measure how the population of these French isolates was structured in a global context. A

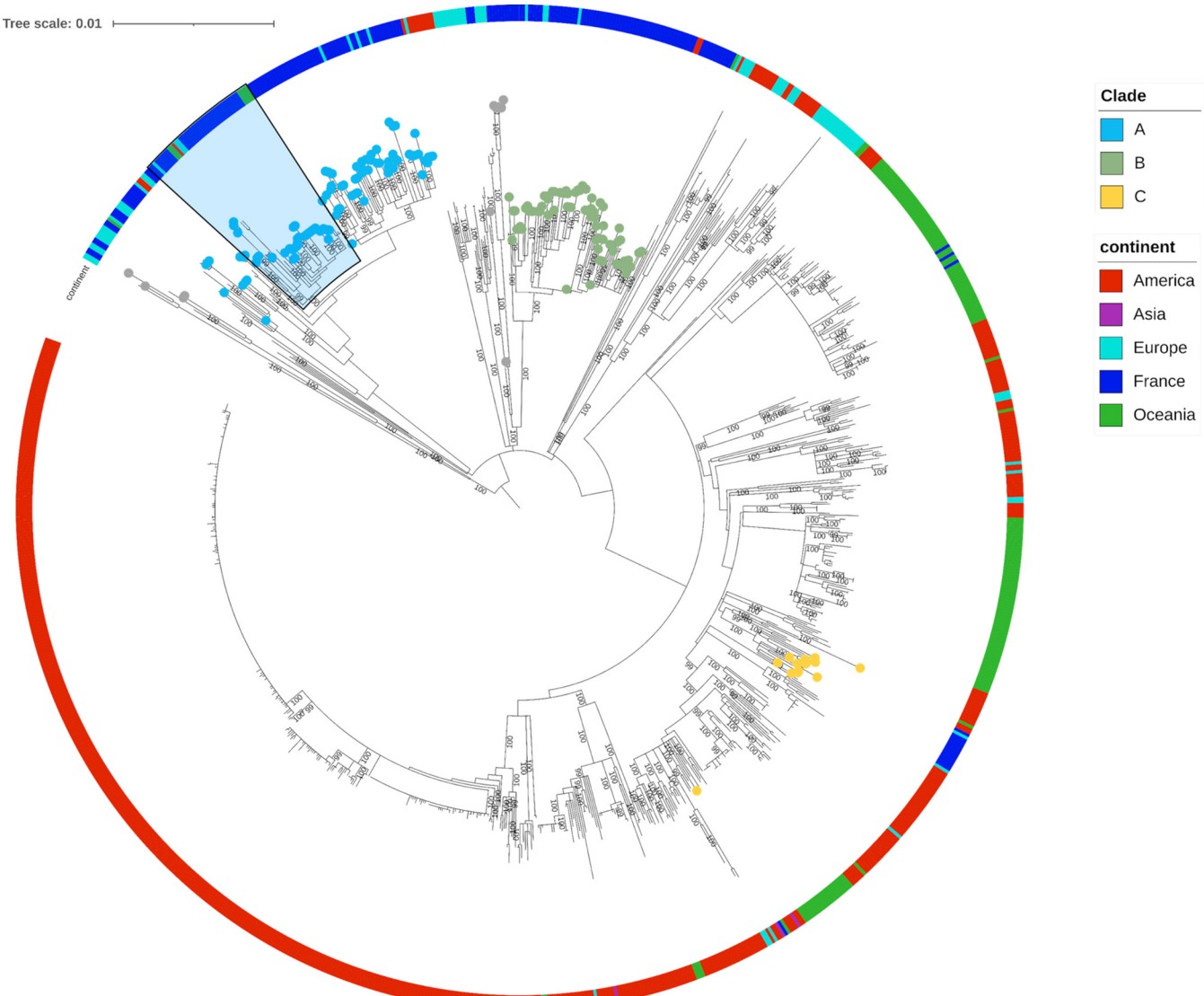

**FIG 7** Global phylogeny of *M. avium* subsp. *paratuberculosis*. A maximum likelihood SNP-based phylogenetic tree of all type C genomes included in this study was inferred using IQ-Tree (TVM+F model). PICSAR77 was used as the reference, and homoplasic sites were removed. The tree was midpoint rooted. The scale of the tree is by site substitution. Previously described phylogenetic clades are shown as leaves on the tree. The outer circle was colored according to continent of origin of isolates. The blue-shaded section indicates the monophyletic group carrying the mutation in the *gyrB* gene.

maximum likelihood phylogeny was inferred based on 9,902 nonhomoplasic SNPs, including 1,025 type C French and international *M. avium* subsp. *paratuberculosis* isolates (Fig. 7 and Table S1, Data Sets 1, 2, 3, and 4). Overall, most of the isolates group by country and we can distinguish major lineages, including isolates predominantly from France, Oceania, or the Americas. Of note, European and American isolates are distributed all along the phylogeny. Looking more specifically at the population of French isolates, clade A includes 13 isolates from European countries, seven from Oceania and three from the Americas. Clade B includes five isolates from European countries and one isolate from the Americas, while clade C includes isolates from Europe. Moreover, the two minor lineages identified in the French diversity are intertwined with isolates from European countries. In addition, it is important to note that strains in clade C are genetically related to strains from the Americas. This suggests that the French *M. avium* subsp. *paratuberculosis* diversity is interconnected with the global *M. avium* subsp. *paratuberculosis* diversity through numerous imports and exports, among which introductions from only three different *M. avium* subsp.

*paratuberculosis* genetic backgrounds gave rise to the majority of isolates sampled in western France (i.e., clades A, B, and C).

Interestingly, the nonsynonymous mutation of the *gyrB* gene at position 1,124 detected in 32 French *M. avium* subsp. *paratuberculosis* genomes (Fig. 3) of the A1 clade was also found in genomes enrolled in this study, including 5 Australian strains, 2 Northern Ireland strains, 1 Canadian strain, 1 Spanish strain, and 1 New Zealand strain (Table S6). These genomes cluster within the monophylogenetic group in the A1 clade indicated in the blue-shaded section in Fig. 7.

## DISCUSSION

The growing number of studies based on the analysis of the whole genome show that these approaches are essential to study the biology of *M. avium* subsp. *paratuberculosis*, a clonal species, and to answer the key questions on the phylodynamics of transmission of this major pathogen in animal health, which conventional molecular typing techniques cannot address. Indeed, these low-resolution techniques are not suitable for precisely measuring the small genetic distances between strains. In this study, we used WGS on a large panel of clinical *M. avium* subsp. *paratuberculosis* isolates sampled from longitudinal follow-up studies of naturally infected cattle. These Normande and Holstein cattle had been selected to establish traits of genetic susceptibility to *M. avium* subsp. *paratuberculosis* in previous studies (18, 24). This sampling of clinical isolates was also selected to be representative of strains present on dairy farms in France. Samples were collected over 4 years and included, in some cases, several per farm as well as several per animal. Our results have thus provided for the first time a very precise image of the genomic characteristics within the French *M. avium* subsp. *paratuberculosis* population.

The initial results of this work underlined how *M. avium* subsp. *paratuberculosis* is a clonal species. These results are in agreement with previous work carried out on the complete genome of *M. avium* subsp. *paratuberculosis*, where the core genome represented 80.3% (14). Even if the S-type *M. avium* subsp. *paratuberculosis* genomes had been added to our panel of French *M. avium* subsp. *paratuberculosis* sequences, all of which are C type, the core genome would still be 98%. This closed pangenome with such a high proportion of core genes is very different from *M. avium* subsp. *hominissuis*, from which evolved the *M. avium* subsp. *paratuberculosis* pathogen (25). Indeed, *M. avium* subsp. *hominissuis* has an open pangenome, as shown by previous studies, which agree with our analysis showing a core genome of only 40%. Unlike *M. avium* subsp. *paratuberculosis*, *Mycobacterium bovis*, another major mycobacterial pathogen in cattle but circulating in a multihost system, has genome characteristics of an open pangenome based on the Roary analysis of 70 draft genomes (26). Recently, a study showed that fragmentation from incomplete genomes severely impacted pangenomic analyses and that *Mycobacterium bovis* actually possessed a closed pangenome (27). Our study agrees with the results of a previous report on 316 *M. avium* subsp. *paratuberculosis* draft genomes that predicted a very small accessory genome (28). It is important to consider that draft genomes introduce a number of problems related to the quality of sequencing data, misassembly, and annotation of fragmentated genomes. All of these problems can influence the size of the pangenome and artificially inflate the size of the accessory genome by the presence of false-positive genes (mainly annotated as "hypothetical protein" genes) and by the absence of certain genes in the draft genomes (27, 29–31). To avoid as much as possible these biases in our pangenomic analysis, we searched for and excluded redundant gene clusters. The remaining accessory genes were searched in the draft and complete genomes, revealing that the majority had undergone truncation at the contig ends. This result suggests that in our panel of four complete genomes from France and 12 complete genomes from other countries, there is no gene content variation within *M. avium* subsp. *paratuberculosis*. According to some reports, a bacterial species whose population is highly clonal with a closed pangenome is theoretically more successful in colonizing stable environments

such as livestock (29). This is particularly well documented for plant pathogens evolving in agricultural ecosystems composed of genetically depauperate populations of plants grown at high density and on large spatial scales (32). The particular architecture of these agro-ecosystems facilitates the emergence, evolution, and dispersal of pathogens for which both specialization and speciation have been observed. In a context where breeding is characterized by the general uniformity of the host, within breeds, *M. avium* subsp. *paratuberculosis* seems to have evolved toward optimal efficiency.

The advent of next-generation sequencing (NGS) not only has helped establish relationships down to the strain level but has also revealed genome-wide differences between strains (33). This is the first study to use WGS to examine the population structure of *M. avium* subsp. *paratuberculosis* in France. The sampling was mainly intended to cover the three regions of France where dairy farms are concentrated and to target the two major breeds that had been the subject of a longitudinal follow-up study (18) examining the genetic susceptibility traits of JD. The phylogeny data clearly indicate the introduction of different genotypes in two main waves, which are clades A and B. Our results from the assembled strain panel suggest a rapid phase of *M. avium* subsp. *paratuberculosis* strain diversification from divergent genetic backgrounds followed by population stabilization within the clades. By integrating strains from 2003, we were able to discern that these genotypes have been circulating in France since at least that date, and it would be interesting to sequence older documented isolates to learn more about the history of *M. avium* subsp. *paratuberculosis* introduction in these regions. The valuable work of Richards et al. shows that these scenarios can be specified with a panel of organized and documented dairy herds over a period of 7 years (28).

Clade C is divergent from the other French clades. The strains belonging to clade C are genetically related ($\leq$100 SNPs) to the strains isolated from the Americas. Fu's $F_s$ test showed that this clade has potentially undergone a bottleneck recently. As it has been suggested that *M. avium* subsp. *paratuberculosis* is mainly transmitted through the animal trade (34–36), like other pathogens (37), it is possible to assume that this bottleneck could correspond to the introduction of this clade in France through the animal trade. However, it is important to note that the low number of isolates in clade C does not enable insights on the clade origin.

Our observations regarding the *gyrB* mutation detected both in the genomes of the French strains and also in the genomes of strains originating from different countries, clustered in the same monophyletic group, suggest that these isolates were derived clonally from the same common ancestor that had acquired this mutation.

Recent studies have focused on the reconstruction of *M. avium* subsp. *paratuberculosis* transmission chains in farms with endemic infection (16, 28, 38, 39). The remarkable study by Nigsch et al. (39) demonstrates that the level of precision WGS data can provide, along with a carefully sampled and informed panel of strains, the ability to make it possible to trace different episodes of infection with genomically distinct clusters, depending on the type of closed or open breeding herds. In our study, we did not have these cases of closed versus open breeding. Moreover, a previous study had mapped the movements of animals (purchase and sale) in this region of France (34), thus demonstrating the intensity of the exchanges, which could explain why we found no association between genotypes and geographical area. Nevertheless, for some farms from which several isolates were analyzed, we observed several genotypes within the same herd, which suggests a different introduction, probably linked to the purchase of animals. Considering the study design by Richards et al. (28), it would be wise to refine more targeted sampling of a few closed versus open farms and combine animal data, including their movements, to better understand the history of the infection.

In our study, the strains were obtained from only affected animals. It would be important for future studies using WGS for epidemiologic evaluation of the *M. avium* subsp. *paratuberculosis* transmission dynamic and to better understand the biology of this pathogen to analyze strains from animals infected at different stages of the

disease, including from asymptomatic animals. The advent of NGS sequencing applied to *M. avium* subsp. *paratuberculosis* epidemiology has clearly demonstrated that animals can be infected by several *M. avium* subsp. *paratuberculosis* genotypes (28, 39–41). Recent studies have even demonstrated the incidence of a mixed and simultaneous infection of several strains, including up to five isolates comprising different genotypes in a single dairy cow (39). In our study, we were able to observe mixed infections, with at least one example of two different genotypes in one animal. Therefore, coinfection should be considered in future studies to better understand *M. avium* subsp. *paratuberculosis* transmission. Several possibilities can lead to the isolation of different genotypes from the same animal. Mixed infections may be due to strain diversification over time from mutations, even though the mutation rate is very low for *M. avium* subsp. *paratuberculosis,* at around 0.25 to 0.5 SNP per genome per year (12, 39). Mixed infections can also result from multiple introductions of various genotypes over time or even exposure of animals to several genotypes. In this study, mixed infections are defined on the basis of WGS. We did not consider MLVA or MLSSR typing to define the genotype of the strains. Indeed, these markers do not seem to be usable to establish phylogenetic links. Among our panel of isolates, we were able to confirm the homoplastic nature of these markers (10, 15), which showed several strains identical in sequence but with different MLVA and MLSSR profiles and, conversely, strains separated by several hundred SNPs yet with identical MLVA and MLSSR profiles.

**Conclusions.** This work provides for the first time the genomic characteristics of the *M. avium* subsp. *paratuberculosis* strain population circulating in France. Because the C type of *M. avium* subsp. *paratuberculosis* is highly clonal, genome sequence analysis is critical to obtain the necessary resolution of these genetically related strains, which we could divide into defined clades. Furthermore, this approach demonstrated the presence of mixed infections in herds and in animals. This work should open up prospects for studying the transmission dynamics of *M. avium* subsp. *paratuberculosis* with appropriate sampling and by combining animal data (history of cattle, filiation, place of birth, purchase and/or sale, information on breeding farms, etc.) with epidemiological and health data. These results should make it possible to quantify the percentage of *M. avium* subsp. *paratuberculosis* transmission due to animal exchange (market) compared to other transmission routes, including environmental sources, where *M. avium* subsp. *paratuberculosis* can survive, disseminate, and evolve. The perspectives from this work will help to develop innovative tools for managing farms intended for the epidemio-surveillance of paratuberculosis, the optimization of control plans, and the management of animal movements.

## MATERIALS AND METHODS

**Strains and genomes.** In this study, five data sets were used, each is described below.

**(i) Data Set 1.** Data Set 1 included an original panel of 182 clinical isolates of *M. avium* subsp. *paratuberculosis* from a longitudinal study in dairy cattle herds in France (Fig. 1) previously described. One study targeting these herds with high JD seroprevalence in France has accurately described fecal shedding patterns relating to individual animal characteristics (age, breed, and parity) and serological patterns (18). A second study aimed at identifying genomic differences between animals with high fecal shedding profiles and uninfected animals in positive herds. Feces from animals with JD clinical signs were cultured to isolate *M. avium* subsp. *paratuberculosis* (19). The culture and sequencing of the 182 isolates collected over a period of 4 years (2014 to 2018) are described elsewhere (20), except for PICSAR126, which was excluded in the present study because too many clade-specific SNPs were ambiguously assigned.

**(ii) Data Set 2.** To investigate French *M. avium* subsp. *paratuberculosis* population structure changes over a long period, 10 *M. avium* subsp. *paratuberculosis* strains collected in 2003 were sequenced, and the sequences were deposited in the NCBI public database.

**(iii) Data Set 3.** To broaden comparative genomic analysis and provide a global vision of *M. avium* subsp. *paratuberculosis* phylogeny, we included all whole-genome sequences available for *M. avium* subsp. *paratuberculosis* in NCBI SRA, GenBank, and ENA, for which all information regarding country of origin and year of isolation was available. This added 826 genomes, which were downloaded in September 2021.

**(iv) Data Set 4.** For synteny studies and genetic characterizations, the 12 complete genomes recently available for *M. avium* subsp. *paratuberculosis* were downloaded from NCBI RefSeq in August 2021.

**(v) Data Set 5.** In addition, a total of 123 *Mycobacterium avium* subsp. *hominissuis* genomes, including 13 complete and 101 draft genomes available at NCBI, were downloaded in September 2021 to compare the pangenomic analyses. Nine genomes were excluded from the analysis based on the contigs and gene

numbers, leaving us with 114 genomes available for analysis (see the "Pangenome analysis" section below).

Data Sets 1 to 5 are described in Table S1 in the supplemental material.

**Pangenome analysis.** Sequences from 16 complete genomes (4 from Data Set 1 and 12 from Data Set 4 in Table S1) and 137 draft genomes remaining from Data Set 1 (Table S1) were annotated using Prokka v.1.14.6 (42) with *M. avium* subsp. *paratuberculosis* K-10 as the reference annotation strain (43). Prodigal (44) was pretrained using the complete genome sequence of strain K-10. The same procedure was used to annotate the sequences of the 13 complete and 110 draft *M. avium* subsp. *hominissuis* genomes from Data Set 5 (Table S1). Next, panaroo-qc was used to check the quality of *M. avium* subsp. *hominissuis* draft genomes. Outlier genomes in terms of number of contigs and number of genes were excluded from the analysis, which excluded 9 genomes and left 114 genomes for analysis. Panaroo (v.1.2.8) (31) was used to delineate the pangenomes of both subspecies independently. The pipeline generates initial gene clusters using a greedy incremental clustering approach that processes sequences based on length (as implemented in CD-HIT). Next, neighborhood information is generated by constructing a graph of the pangenome where nodes are gene clusters and edges connect genes adjacent on contigs. The graph and neighborhood information are then used to identify and merge fragmented or mistranslated genes and identify genes missed by the gene-calling algorithm. To detect any redundant gene clusters that can inflate the estimated pangenome's size, consensus sequences of all gene clusters constituting the pangenome were aligned to a complete genome—i.e., FDAARGOS_305 (also known as the K-10 strain) (NCBI accession no. NZ_CP022095.2) (45) for *M. avium* subsp. *paratuberculosis* and 104 (NCBI accession NC_008595.1) for *M. avium* subsp. *hominissuis*—using blastn (46) with an identity cutoff of 90% and a similarity E value cutoff of $1 \times 10^{-10}$. Redundant gene clusters were labeled using the R IRanges (v.2.28.0) package (47) if the best blastn result from one cluster overlapped another cluster by at least 75%, creating a matching network. Only the most represented gene cluster in this network was retained, with the others removed from the pangenome.

The pangenome matrix was filtered to select core genes and build core gene alignments using the following three exclusion criteria:

1. Gene families not present in all samples were excluded.
2. Gene families in multicopy were excluded.
3. Gene families with partial sequence from one assembly were excluded.

Concatenated core gene alignment was the first test for the presence of recombinant sequences using the pairwise homoplasy index (PHI) test in SplitTree (v.4.18.0) (48), with a significance threshold of $P \leq 0.05$. An ML tree was inferred based on the core gene alignments using IQ-Tree (v.1.6.9) with 100 bootstraps as pseudoreplicates. Selection of the best-fit model of evolution, K3Pu+F+I, was performed in IQ-Tree based on the lowest Bayesian information criterion. The phylogenetic tree was visualized using Interactive Tree of Life. Python script roary_plots.py (https://github.com/sanger-pathogens/Roary/blob/master/contrib/roary_plots/roary_plots.py) was used to plot core gene phylogeny beside the pangenome matrix.

Core and pangenome curves were plotted using a modified version of Python script from https://github.com/rotheconrad/00_Pangenome_Analysis/blob/master/04b_Pangenome_Calculate_Model_Plot.py with 1,000 permutations. The openness of the pangenome was tested using home-made Python script by fitting the Heaps' law model according to Tettelin et al. (22) to the mean curve of the new genes using the curve_fit function of the scipy.optimize package (49).

**Single-molecule real-time sequencing and assembly.** Four genomes of French *M. avium* subsp. *paratuberculosis* strains (Data Set 1 in Table S1) were selected on the basis of phylogeny clustering to create complete genomes of reference for *M. avium* subsp. *paratuberculosis* strains currently circulating in France: strains PICSAR51, PICSAR68, PICSAR77, and PICSAR142.

*M. avium* subsp. *paratuberculosis* strains were isolated from fecal samples on slopes of Herrold's egg yolk medium containing mycobactin J, amphotericin B, and nalidixic acid (Becton Dickinson, Le Pont de Claix, France) according to the method of Whipple et al. (50). Then, *M. avium* subsp. *paratuberculosis* subcultures were propagated from colonies in Middlebrook 7H9 medium with 0.5% (vol/vol) glycerol, 10% (vol/vol) Middlebrook oleic acid-albumin-dextrose-catalase (OADC) enrichment medium (Becton, Dickinson, Oxford, Oxfordshire, United Kingdom), and 2 $\mu$g/mL of mycobactin J (IdVet, France) at 37°C. The strains were tested for the presence of the *M. avium* subsp. *paratuberculosis* sequence IS*900* and genotyped using the MLVA method (10). DNA was extracted using the NucleoBond AXG 100 DNA kit (Macherey-Nagel, Germany) according to the manufacturer's instructions. DNA was quantified using a Qubit double-stranded DNA (dsDNA) BR assay kit and Qubit 4 fluorometer (Invitrogen, Thermo Fisher Scientific). Library preparation and single-molecule real-time (SMRT) sequencing were performed by GenoScreen (Lille, France) on the PacBio Sequel platform.

Long reads shorter than 1,000 bp were removed. A hybrid assembly was performed with the remaining long reads and short reads for the samples of PICSAR51, PICSAR68, PICSAR77, and PICSAR142 using Unicycler (v.0.4.8) (51) and Flye (v.2.8.3-b1695) (52) with default parameters. Briefly, Unicycler performs a SPAdes (v.3.14.1) (53) assembly of Illumina short reads and then scaffolds the assembly graph using long reads. Unicycler polishes its final assembly with short reads and Pilon (v.1.24) (54) to reduce the rate of small base-level errors. This polishing step is rerun until 2 runs produced no further corrections. If both assemblers were unable to produce circular genomes, then the least-fragmented assembly was selected for each genome. To resolve genome structure and ascertain that these genomes were complete, contig extremities were extracted and aligned first to the assembly to identify any overlap and second to two complete genomes (K-10 and MAP4) to identify any assembly gap. Assembly gaps were found by

alignment of contig extremities on complete genomes to locate and extract regions. These regions were aligned back to the assembly using BLAST (46) to get gap sizes. Short reads and long reads were mapped to the sequence containing the gaps with Burrows-Wheeler Aligner mem (BWA-MEM) (v.0.7.17-r1188) (http://arxiv.org/abs/1303.3997) and minimap2 (v2.21-r1071) (55) to check for the presence of gap sequence in the sample genome and if so extract them to perform a *de novo* assembly with Unicycler. The assembled contig was aligned to genome assembly in order to insert the missing sequence. Genomes were rotated to start at the *dnaA* gene using the Circlator (v.1.5.5) (56) fixstart command. Genomes were annotated using Prokka (v.1.14.6) (42) with *M. avium* subsp. *paratuberculosis* K-10 as the reference annotation strain. Prodigal (44) was pretrained using the complete genome sequence of strain K-10.

All assemblies were aligned using progressiveMauve (v.2.14.0) (57) and visualized with Mauve (v snapshot_2015-02-13 build 0) (58) in order to highlight any misassembly resulting from the assembly protocol.

**Variant calling and phylogenetic analysis.** For complete genomes (Table S1, Data Set 4), paired-end reads (250 bp) were simulated using ART software (v.2.5.8) (59) based on the HiSeq 2500 platform. Illumina reads for Data Set 3 (Table S1) were downloaded from NCBI's SRA database using fasterq-dump (v.2.10.1). Raw sequencing data were processed as previously described (20). Briefly, short reads were trimmed using fastp v.0.20.1 and a Phred scale base quality threshold of Q30. Kraken2 v.2.0.9-beta with a bacterial database was used to identify potential contamination. Only short reads assigned to the *Mycobacteriaceae* family were retained in the analyses.

To avoid potential alignment error from underlying structural variation compared to the reference genome and to define phylogenetic relationships as precisely as possible (60), reads were mapped to a local reference from the panel, namely, PICSAR77 (GenBank accession no. CP091844 [Table S1, Data Set 1), using BWA-MEM (v.0.7.17-r1188). SNPs were detected with GNU parallel (https://zenodo.org/record/3841377) and FreeBayes (v.1.3.2-dirty) (https://arxiv.org/abs/1207.3907) with –minqual and –minmapq options set to 20 and 30, respectively, for each strain. SNPs were filtered out based on four criteria: (i) the quality of the SNP is more than 300, (ii) at least 2 reads support the variant on the forward and reverse strands, (iii) at least 2 reads support the variant placed on the left and right, and (iv) there is a mean mapping quality at a site of 60 to avoid read mapping error. VCF files from each strain were pulled together to list all detected positions in the population, and variant recall was performed only on these positions. Recall VCF files were merged to obtain a global VCF encompassing all strains. A global VCF file was functionally annotated with the snpEff tool (v.4.5covid19) (61). In-house script was developed to build the SNP concatenate. SNPs were classified at positions where $\geq$ 90% of reads support the alternative allele (ALT). When reads' support values for the ALT were $>$10% and $<$90% (defined as heterozygous SNP) or depth was lower than 4, an ambiguous character, *N*, was called in the alignment. Positions with an ALT reads' support of $\leq$10% were classified as "reference." Strains for which half of the clade-specific SNPs (see the phylogeny sections) were assigned to an ambiguous character, *N*, were excluded from the analysis. Only PICSAR126 was excluded, with 68% of ambiguous positions. Homoplasic positions in the alignment were removed using HomoplasyFinder (62).

Homoplasy-removed SNP alignments were used to generate Bayesian and maximum likelihood (ML) trees. IQ-Tree (v.1.6.9) (63) was used to infer the ML tree with 100 bootstraps as pseudoreplicates. Selection of the best-fit model of evolution, K3Pu+F or TVM+F, was performed in IQ-Tree based on the lowest Bayesian information criterion. MrBayes (64) was used to infer the Bayesian tree, with 1,500,000 generations and 25% burn-in with the GTR+G+I substitution model. Trees were midpoint rooted. The phylogenetic tree was visualized using Interactive Tree of Life (65).

Population structure was validated using a Bayesian clustering/assignment approach, as implemented in fastBAPS software (66) using the homoplasy-removed SNP alignments and ML phylogeny as the input.

A regression of the sampling date (Table S1, Data Sets 1 and 2) against the root-to-tip genetic distances performed with Tempest (67) showed that the data set contains insufficient temporal signal for a molecular clock analysis (data not shown).

**Quantification of population growth/expansion.** To detect departure from a standard neutral model, a series of tests was used on the French population individually. DNASP v.6 (68) was used to perform Tajima's *D* (69), Fu's $F_s$ (70), and Fu's and Li's *D\** and *F\** tests (71). This tool provides *P* values based on a coalescent simulation algorithm. (A total of 10,000 simulation runs were performed.) For the Tajima's *D* test, the *P* values represent the probability that the simulated estimate is less than the observed value and is less than or equal to the observed value for the rest of the tests. Rejection of these tests may be caused by violation of any of the assumptions in the null hypotheses (neutrality, constant size, panmixia, or no recombination).

**Phylogeny-trait correlation.** Correlation between phylogenetic tree structure and trait values for each isolate were assessed using BaTS (Bayesian Tips Significance Testing) (72). This program tests the null hypothesis of no correlation between phylogeny and traits by performing randomization tests. Trait values were randomized 100 times to yield a null distribution for hypothesis testing. A correlation was unambiguously positive if both the association index (AI) and the parsimony score (PS) statistics rejected the null hypothesis with a *P* value of $\leq$ 0.01. Phylogenetic uncertainty was considered by using the set of tree topologies estimated by Bayesian phylogenetic inference (MrBayes).

**Comparison between phylogenetic and geographical distances.** Pairwise phylogenetic distances between isolates in each clade were calculated with the "cophenetic.phylo" function implemented in the R package "ape" (v.5.6.1) (73). Redundant isolates (i.e., the same farm, same sampling date, and same genetic strain) were removed from each clade. Pairwise geographic distances between isolates were calculated using GPS coordinates of the birth herds or the municipality, when available, with the help of

the "haversine_distances" function implemented in the python package scikit-learn (v1.0) (https://jmlr .csail.mit.edu/papers/v12/pedregosa11a.html). A LOESS (locally estimated scatterplot smoothing) regression and Spearman correlation test were applied to assess the strength of the relationship between phylogenetic and geographic distances.

**Molecular typing.** Multilocus variable-number tandem repeat (VNTR) analysis (MLVA) typing using a combination of VNTR and mycobacterial interspersed repetitive-unit (MIRU) markers was carried out as described elsewhere (10). Multilocus sequence types (MLSTs) based on 11 short sequence repeat (SSR) markers (MLSSR) (74) were deduced from the genomic sequences. The allocation of profiles was generated via the MAC-INMV-SSR database (http://mac-inmv.tours.inra.fr/index.php?lang=fr) (75).

**Cartography.** The maps illustrating the distribution of isolates sampled in the west of France with information on sampling number, cattle breeds (Fig. 1) and typing results (Fig. S3) were generated using MicroReact (v.202) (76).

**Data availability.** The sequencing data for 10 *M. avium* subsp. *paratuberculosis* strains collected in 2003 were sequenced and deposited in the NCBI public database under Bioproject no. PRJNA794728. The genomes assembled in this study have been deposited in the NCBI GenBank database under accession no. CP091842 to CP091845 (Table S1, Data Set 1).

## SUPPLEMENTAL MATERIAL

Supplemental material is available online only.

**SUPPLEMENTAL FILE 1**, PDF file, 0.3 MB.
**SUPPLEMENTAL FILE 2**, XLSX file, 0.1 MB.
**SUPPLEMENTAL FILE 3**, XLSX file, 0.1 MB.
**SUPPLEMENTAL FILE 4**, XLSX file, 0.1 MB.
**SUPPLEMENTAL FILE 5**, XLSX file, 0.6 MB.
**SUPPLEMENTAL FILE 6**, XLSX file, 0.02 MB.
**SUPPLEMENTAL FILE 7**, XLSX file, 0.1 MB.

## ACKNOWLEDGMENTS

We are grateful to the INRAE MIGALE bioinformatics facility (MIGALE, INRAE, 2020, Migale Bioinformatics Facility [https://doi.org/10.15454/1.5572390655343293E12]) for providing help and/or computing and/or storage resources. We thank Aurore Davergne, David Ngwa-Mbot, and Laurent Journaux for providing technical support in the project GISA-PICSAR.

Conceptualization, F.B. and C.C.; Data Curation and Formal Analysis, C.C., J.T., T.C., and J.P.B.; Funding Acquisition, F.B., L.S., and C.F.; Resources, T.C., A.D., A.J., and C.F.; Software, C.C.; Project Administration and Supervision, F.B.; Writing – Original Draft, F.B., C.C., and J.P.B.; Writing – Review & Editing, F.B., C.C., J.T., and J.P.B. All coauthors commented on and approved the manuscript.

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
