## [Reviewer comments · Microbiology Spectrum]

Microbiology Spectrum

Genetic features of *Mycobacterium avium* subsp. *paratuberculosis* strains circulating in the West of France deciphered by Whole-Genome Sequencing

Cyril Conde, Julien Thézé, Thierry Cochard, Marie-Noëlle Rossignol, Christine Fourichon, Arnaud Delafosse, Alain Joly, Raphael Guatteo, Laurent Schibler, John Bannantine, and Franck Biet

Corresponding Author(s): Franck Biet, INRAE centre val de Loire

Review Timeline:

Submission Date:	August 25, 2022
Editorial Decision:	October 30, 2022
Revision Received:	November 9, 2022
Accepted:	November 10, 2022

Editor: Sacha Pidot

Reviewer(s): Disclosure of reviewer identity is with reference to reviewer comments included in decision letter(s). The following individuals involved in review of your submission have agreed to reveal their identity: Sheldon T Brown (Reviewer #2)

Transaction Report:

DOI: <https://doi.org/10.1128/spectrum.03392-22>

October 30, 2022

Dr. Franck Biet
INRAE centre val de Loire
ISP
Route de Crotelle
Nouzilly, 37 F37380
France

Re: Spectrum03392-22 (Genetic features of *Mycobacterium avium* subsp. *paratuberculosis* strains circulating in the West of France deciphered by Whole-Genome Sequencing)

Dear Dr. Franck Biet:

Thank you for your patience with the review process. The reviewers were very positive about your work and praised its important contribution to our understanding of MAP infection dynamics. Please see the reviewers comments (below and attached) for minor modifications to the manuscript before it can be accepted for publication.

Link Not Available

Sincerely,

Sacha Pidot

Journals Department
Reviewer comments:

Reviewer #1 (Comments for the Author):

This manuscript reveals new insights into Map strains circulating in Western France. I have only one minor suggestion: Animal 85, not 86, cited in line 244, page 11, show mixed infections as presented on table S4.

Reviewer #2 (Comments for the Author):

See attachment.

Staff Comments:

Preparing Revision Guidelines

Please return the manuscript within 60 days; if you cannot complete the modification within this time period, please contact me. If you do not wish to modify the manuscript and prefer to submit it to another journal, please notify me of your decision immediately so that the manuscript may be formally withdrawn from consideration by Microbiology Spectrum.

This manuscript reveals new insights into Map strains circulating in Western France. I have only one minor suggestion:

Animal 85, not 86, cited in line 244, page 11, show mixed infections as presented on table S4.

MAP WGS France Summary

Whole genome sequencing (WGS) of 200 isolates of *M. avium* sp. *paratuberculosis* (MAP) was used to analyze the genetic characteristics of MAP isolated from infected cattle in France. Results were compared with available MAP WGS findings from around the world.

Major results

A closed pangenome for MAP as compared to *M. avium* sp. *hominissuis* (MAH) was confirmed with the preponderance of genes belonging to the core genome demonstrating the highly clonal nature of MAP. Only C type MAP was found, which phylogenetically distributed into 3 closely related clades. No cattle breed selectivity was apparent. Mixed infection within herds and individual animals was identified. Non-homoplastic phylogeny based upon single-nucleotide-polymorphisms SNP alignment demonstrated the homoplastic nature of variable-number-tandem-repeat (VNTR) and short-sequence-repeat (SSR) genotyping methods. WGS thus provides resolves strain relatedness at a more granular level than traditional methods for strain identification.

Significance

The findings strongly imply that variations in MAP infection within dairy herds result primarily from herd introduction through animal trade. This is the first analysis of MAP by WGS in France, is a substantial contribution to extant databases of MAP WGS, solidifies WGS as the most reliable method of MAP strain identification, and provides a foundation for future epidemiologic investigation of this economically important disease.

Comments:

The manuscript is well written. The sequence of presentation has been carefully considered enabling the reader to work through complex methodology and analysis.

The supplemental tables and figures are all helpful and informative for readers seeking more detail.

A limitation of the study, not expressly alluded to in the discussion is that the analysis is based upon herd sampling that was intended to evaluate the dynamics of shedding in herds known to be infected. The isolates obtained are thus skewed more toward animals that are likely to be heavily infected. The relative frequency, for example, of type C clades identified may therefore not be reflective of strain virulence, but rather be a convenience sample of strain amplification within chronically infected herds. While there is no claim of epidemiological precision in the manuscript, this is an important limitation that should inform the design of future studies using WGS for epidemiologic evaluation of MAP transmission dynamic and interventional efforts to reduce herd prevalence.

Dear Editor,

Author responses point-by-point to the issues raised by the reviewers on the Spectrum03392-22 entitled "Genetic features of *Mycobacterium avium* subsp. *paratuberculosis* strains circulating in the West of France deciphered by Whole-Genome Sequencing " by Cyril Conde *et al.*

Reviewers' comments:

Reviewer #1:

This manuscript reveals new insights into Map strains circulating in Western France. I have only one minor suggestion:

Animal 85, not 86, cited in line 244, page 11, show mixed infections as presented on table S4.

Authors' response:

Thank you for the comment.

There was indeed an error concerning animal 85 between the text and the table S4, thank you for pointing out this detail. This phrase L240 has been corrected and modified for clarity as follows: At the animal level the same phenomenon, i.e. clonal infection, was observed with 3 identical isolates sampled over a 2-month period from animal 85 (see Table S4, animal 85, herd 18).

L242-L244 the herd number have been indicated in this version

L249 animal number has been corrected, 85 to 86 in this revised version

Reviewer #2

Comments:

The manuscript is well written. The sequence of presentation has been carefully considered enabling the reader to work through complex methodology and analysis.

The supplemental tables and figures are all helpful and informative for readers seeking more detail.

A limitation of the study, not expressly alluded to in the discussion is that the analysis is based upon herd sampling that was intended to evaluate the dynamics of shedding in herds known to be infected. The isolates obtained are thus skewed more toward animals that are likely to be heavily infected. The relative frequency, for example, of type C clades identified may therefore not be reflective of strain virulence, but rather be a convenience sample of strain amplification within chronically infected herds. While there is no claim of epidemiological precision in the manuscript, this is an important limitation that should inform the design of future studies using WGS for epidemiologic evaluation of MAP transmission dynamic and interventional efforts to reduce herd prevalence.

Authors' response:

Thank you for these comments.

We agree, the reviewer raises here a very important point, our sampling came from infected and affected animals. So, we have only looked at the tip of the iceberg. To study the complex

epidemiology of the JD it would be important to include strains isolated from animals infected at different stages of the disease including the asymptomatic stage.

We have added a sentence L409 mentioning this limitation in the discussion of this revised version as follow: In our study the strains were obtained from only affected animals. It would be important for future studies using WGS for epidemiologic evaluation of *Map* transmission dynamic and to better understand the biology of this pathogen, to analyze strains from animals infected at different stages of the disease including from asymptomatic animals.

November 10, 2022

Dr. Franck Biet
INRAE centre val de Loire
ISP
Route de Crotelle
Nouzilly, 37 F37380
France

Re: Spectrum03392-22R1 (Genetic features of *Mycobacterium avium* subsp. *paratuberculosis* strains circulating in the West of France deciphered by Whole-Genome Sequencing)

Dear Dr. Franck Biet:

Your manuscript has been accepted, and I am forwarding it to the ASM Journals Department for publication. You will be notified when your proofs are ready to be viewed.

Sincerely,

Sacha Pidot
Editor, Microbiology Spectrum

Journals Department
Supplemental Material: Accept
Supplemental file 1: Accept
Supplemental Material: Accept
Supplemental Material: Accept
Supplemental Material: Accept
Supplemental Material: Accept
Supplemental Material: Accept